# Recommendation Systems with Distribution-Free Reliability Guarantees

## Abstract

When building recommendation systems, we seek to output a helpful set of items to the user. Under the hood, a ranking model predicts which of two candidate items is better, and we must distill these pairwise comparisons into the user-facing output. However, a learned ranking model is never perfect, so taking its predictions at face value gives no guarantee that the user-facing output is reliable. Building from a pre-trained ranking model, we show how to return a set of items that is rigorously guaranteed to contain mostly good items. Our procedure endows any ranking model with rigorous finite-sample control of the false discovery rate (FDR), regardless of the (unknown) data distribution. Moreover, our calibration algorithm enables the easy and principled integration of multiple objectives in recommender systems. As an example, we show how to optimize for recommendation diversity subject to a user-specified level of FDR control, circumventing the need to specify ad hoc weights of a diversity loss against an accuracy loss. Throughout, we focus on the problem of *learning to rank* a set of possible recommendations, evaluating our methods on the Yahoo! Learning to Rank and MSMarco datasets.

## 1 Introduction

The digitization of all manner of services has introduced recommendation systems into many aspects of our day-to-day lives. In particular, recommendation systems are now being applied to safety-critical domains such as making lifestyle recommendations to patients in healthcare (Hammer et al., 2015; Tran et al., 2021). It is therefore increasingly important that deployed recommender systems do not output recommendations devoid of uncertainty annotations. Meaningful recommendations should come with transparent and reliable statistical assessments. To date, the majority of deployed systems have fallen far short of this desideratum (Covington et al., 2016; Liu et al., 2017; Geyik et al., 2018).

Augmenting recommendation systems with internal tracking of statistical error rates would unlock new capabilities and applications. One such capability is the ability to enforce auxiliary constraints while still guaranteeing a baseline number of high-quality items in each slate of recommendations. For example, we could diversify slates whose quality we are confident in, while leaving lower-confidence slates untouched. Furthermore, the strong guarantees provided by uncertainty quantification are a prerequisite for applying recommendation systems to safety-critical tasks such as medical diagnosis, where a misdiagnosis due to uncertain predictions can be fatal.

### 1.1 Our Goal

In this paper, we develop a method for quantifying uncertainty for the task of learning to rank (L2R). In particular, we consider the setting where we seek to return only items of some quality level, and can return sets of variable size. When returning variable-sized sets, a canonical notion of statistical error is the *false discovery rate* (FDR) (e.g., Efron, 2010). We will focus on this quantity in this work.

Formally, let $\{1, \ldots, K\}$ be the items under consideration, let $Y^* \subset \{1, \ldots, K\}$ be the ground-truth subset of the items that are of acceptable quality, and let $\widehat{S} \subset \{1, \ldots, K\}$ be the set of items returned by the algorithm.

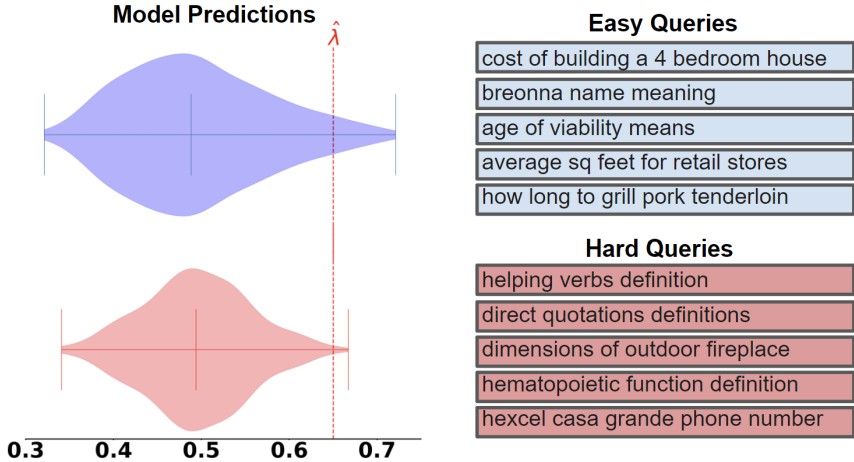

Figure 1: Examples of easy and hard queries. The violin plots show the distribution of document quality scores output by a LambdaRank model for the five queries on the right. For easy queries, the model can distinguish between the qualities of different documents, so the violin has a wide spread, and vice-versa for hard queries.

The false discovery rate of an algorithm is

$$\text{FDR} \;=\; \mathbb{E}\left[\frac{|\widehat{S} \cap Y^*|}{\max(|\widehat{S}|, 1)}\right],$$

and we ask the algorithm to control this quantity at some user-specified level $\alpha$.

In words, controlling for FDR means that the algorithm returns sets that are mostly items of high quality. When queries return large sets, it indicates that the model can confidently identify many high-quality items. Conversely, when queries return small sets, it indicates that the model cannot identify high-quality items with confidence. To control for FDR in recommendation systems, we propose a calibration algorithm that returns set of items with FDR guaranteed to be lower than a user-specified level with high probability; see Figure 1 for an example. This algorithm applies to any L2R model, including neural net models trained with LambdaRank (Burges, 2010). The algorithm comes with finite-sample statistical guarantees whatever the model, and these guarantees enable users to interact with recommendations with confidence.

Rigorously tracking statistical error rates opens the door to further performance improvements within a system. In this work, we will focus on the *diversity* of the items returned—a central goal when designing recommender systems. In particular, we will show how to return sets that are optimized for diversity *while maintaining finite-sample FDR guarantees*. That is, we will seek to approximately solve

$$\max_{\mathcal{D}} \quad \mathbb{E}\Big[\text{Diversity}\big(\mathcal{D}(X)\big)\Big]$$
$$\text{s.t.} \quad \mathbb{P}\Big(\text{FDR}(\mathcal{D}) > \alpha\Big) < \delta.$$

Here, $\mathcal{D}$ is a model that maps an input to a set of returned items, $\widehat{S}$, and Diversity is a user-specified metric for the diversity of a set of items. See Section 2.3 for details. To our knowledge, this is the first such statistical result in the recommendation systems literature.

## 1.2 Related Work

Accurately quantifying model uncertainty has long been a desirable feature in information retrieval systems. Early work simply aimed to score the relevance of each item (Robertson & Jones, 1976; Koren et al., 2009). However, these methods are not calibrated, therefore their output scores can not be interpreted

as probabilities. To remedy this issue one line of work models the problem through a Bayesian lens (Zhu et al., 2009; Freudenthaler et al., 2011; Gopalan et al., 2014; Wang et al., 2018). While these methods can improve upon their uncalibrated counterparts, they must be developed from scratch and require making strong assumptions about user interactions.

Our work aims to quantify the reliability of recommendations in the learning-to-rank setting. Recent work shows that neural learning-to-rank models suffer from poor calibration (Penha & Hauff, 2021), highlighting the importance of our goals.

Our approach is based on recent developments in *conformal prediction* (Vovk et al., 1999; Papadopoulos et al., 2002; Vovk et al., 2005; Angelopoulos & Bates, 2021) and *distribution-free uncertainty quantification* more broadly (Park et al., 2020; Bates et al., 2021a; Angelopoulos et al., 2021). This line of work provides a formal approach to defining set-valued statistical predictions and it has been applied to various learning tasks, such as distribution estimation (Vovk et al., 2020), causal inference (Lei & Candès, 2020; Jin et al., 2021), weakly-supervised data (Cauchois et al., 2022), survival analysis (Candès et al., 2021), design (Fannjiang et al., 2022), model cascades (Fisch et al., 2020; Schuster et al., 2021), the few-shot setting (Fisch et al., 2021), handling dependent data (Chernozhukov et al., 2018; Dunn et al., 2020), and handling or testing distribution shift (Tibshirani et al., 2019; Cauchois et al., 2020; Hu & Lei, 2020; Bates et al., 2021b; Gibbs & Candès, 2021; Vovk, 2021; Podkopaev & Ramdas, 2022). Most closely related to the present work, there have been recent proposals applying conformal prediction to recommender systems. One line of work aims to apply conformal prediction to quantify the uncertainty in predicted ratings (Himabindu et al., 2018; Ayyaz et al., 2018), while another aims to quantify the uncertainty in a set of recommended items when only implicit feedback is available (Kagita et al., 2017; Penha & Hauff, 2021).

Diverging from these proposals, we go beyond conformal prediction and use the more general risk-control framework (Bates et al., 2021a; Angelopoulos et al., 2021). As a result, our work allows recommender systems to be optimized with respect to metrics other than accuracy while maintaining reliability guarantees. While we focus on diversity as a case study (Kunaver & Požrl, 2017), our work is applicable to the broader literature on alternative metrics, including reachability, serendipity and fairness (Singh & Joachims, 2018; Yao & Huang, 2017; Dean et al., 2020; Herlocker et al., 2004; Kaminskas & Bridge, 2016). As prior work has shown, focusing solely on accuracy can harm performance with respect to these alternate metrics (Adomavicius et al., 2013; Nguyen et al., 2014; Fleder & Hosanagar, 2009), underscoring the necessity of designing recommender systems that can be optimized with respect to multiple objectives.

### 1.3 Our Contribution

We introduce a method for calibrating learning-to-rank models to control the false discovery rate. The calibration procedure is supported by finite-sample statistical guarantees that apply to any model and dataset. Controlling the false discovery rate enables downstream tasks like optimizing recommendations for diversity; we explicitly extend our algorithm to produce sets of high diversity that are certified to control the false discovery rate. This concrete example also serves as a template for how to handle desiderata beyond diversity while providing statistical guarantees.

## 2 Methods

We begin by describing the learning-to-rank problem in recommendation systems. We then present a calibration algorithm for controlling the FDR in recommendation systems with provable guarantees.

### 2.1 Learning to Rank

The *learning-to-rank* problem refers to a task where we receive a query from a user and seek to return a list of responses ranked by their relevance. Formally, for any particular query, we have $K \in \mathbb{N}$ possible responses, *i.e.*, items that could be output. Note that $K$ varies per query in our experiments; however, we suppress this dependence in our notation. We also have a list of *features* $X = \left\{ X^{(j)} \right\}_{j=1}^{K}$ in some space $\mathcal{X}$, where $X^{(j)}$ encodes all relevant information about the $j$th response, including any interactions with the user's identity

and query. One can think of the features as being an embedding from a neural network. Furthermore, we have a *ranking* $Y$ that takes values in $\mathcal{Y} = S_K$, the space of permutations on $K$ items, and determines which of the possible responses are most relevant (earlier-ranked items are more relevant). We use the notation $Y^{(j)}$ to refer to the rank of the $j$th response, and $Y^{(i:j)}$ to mean the subvector of $Y$ from index $i$ to index $j$, inclusive. Finally, we have a model $\widehat{\pi}$ that takes the input $X$ and returns a probability $\widehat{\pi}_{i,j}$, where $\widehat{\pi}_{i,j}$ is an estimate of the probability that item $i$ is preferred to item $j$:

$$\widehat{\pi}_{i,j}(X) = \widehat{P}\Big(Y^{(i)} \leq Y^{(j)}\Big).$$

The model is usually trained from data to approximate the mapping from the features to the ranking.

As a motivating example of our setup, the reader can think of a search engine: $K$ is the number of hits for a query, $X^{(j)}$ is the content within the $j$th hit, and $Y$ is the ideal order in which the hits should be presented on the page. Since we do not know which webpages match the user's query in advance, we estimate it with a machine-learning model $\widehat{\pi}$, then select which results to display. In the next section, we propose an algorithm for returning a short list of provably high-quality responses to the user using the machine-learning model.

## 2.2 FDR-Controlling Sets

Based on the output of $\widehat{\pi}$, we seek to output a final set $\widehat{S} \subset \{1, \ldots, K\}$, that contains mostly good items. Formally, we will seek sets that have a low *false discovery proportion*:

$$\text{FDP}(\widehat{S}, Y) = \frac{|\widehat{S} \cap Y^{(m+1:K)}|}{\max(|\widehat{S}|, 1)}.$$

That is, for any prediction $\widehat{S}$, the FDP is the fraction of $\widehat{S}$ that does *not* fall in the top $m$ items. Here, $m$ is a parameter set by the analyst. For example, we might take $m = .2 \cdot K$, the top 20% of items.

### A family of set-valued functions

We next explain how to produce good sets $\widehat{S}$ from the model output. Note that the $\widehat{\pi}_{i,j}$ need not be properly calibrated. Nonetheless, they do encode our model's assessment of the quality of each item. Therefore, we will create sets that include the most promising items, as judged by the model. In particular, we will rank items based on their total quality:

$$s_i(X) = \frac{1}{K-1} \sum_{j \neq i} \widehat{\pi}_{i,j}(X).$$

A larger $s_i$ represents a more promising item; if the model's probabilities were correct, then $s_i$ would be the expected fraction of other items that item $i$ is better than. We consider sets that include only the best items, as judged by the score above:

$$\mathcal{T}_\lambda(X) = \{i : s_i(X) \geq \lambda\},$$

for $\lambda \in [0, 1]$.

### Calibrating $\lambda$ with Learn then Test

We want to find a rule $\mathcal{T}_\lambda$ that has good FDR properties. That is, we want it to control the *risk*:

$$\text{FDR}(\mathcal{T}_\lambda) = \mathbb{E}\left[\text{FDP}(\mathcal{T}_\lambda(X), Y)\right].$$

Using Learn then Test (Angelopoulos et al., 2021), we can select a parameter $\widehat{\lambda}$ that controls the risk

$$P(\text{FDR}(\mathcal{T}_{\widehat{\lambda}}) > \alpha) < \delta, \tag{1}$$

where $\alpha$ and $\delta$ are parameters set by the user.

We next review how to achieve the risk control in (1) with Learn then Test. Conceptually, the algorithm starts with small recommendations that are certain to control the FDR, and progressively grows them until

---

**Algorithm 1** The Learn then Test calibration procedure for L2R

---

**Input:** Calibration data, $(X_i, Y_i)$, $i = 1, \ldots, n$; risk level $\alpha$; error rate $\delta$; underlying predictor $\widehat{\pi}$; step size $d\lambda > 0$.

**Output:** Parameter $\widehat{\lambda}$ for computing RCPS.

1: $\lambda \leftarrow 1$
2: fdp $\leftarrow 1$
3: **while** fdp $\leq \alpha$ **do**
4:     $\lambda \leftarrow \lambda - d\lambda$
5:     **for** $i = 1, \ldots, n$ **do**
6:         $\text{FDP}_i \leftarrow \text{FDP}\big(\mathcal{T}_\lambda(X_i), Y_i\big)$
7:     fdp $\leftarrow \frac{1}{n} \sum_{i=1}^{n} \text{FDP}_i + \sqrt{\frac{1}{2n} \log \frac{1}{\delta}}$   ▷ Can replace with any valid upper-confidence bound on the risk.
8: $\widehat{\lambda} \leftarrow \lambda + d\lambda$   ▷ Backtrack by one because we overshot.

---

the liminal point where making them any bigger would violate the FDR. Each time we grow the set of recommendations, we must calculate a p-value telling us if the FDR is controlled, and stop if that p-value is greater than $\delta$. Normally, calculating many p-values would incur a multiple testing penalty, but this can be avoided using a protocol known as *fixed sequence testing*; see Wiens (2003). First, we consider a discrete set of values, $\Lambda = (.99, .98, \ldots, .01)$. For each $\lambda \in \Lambda$, we consider the null hypothesis:

$$H_{0,\lambda} : \text{FDR}(\mathcal{T}_\lambda) > \alpha.$$

We test this null hypothesis using independent and identically distributed (i.i.d.) calibration data, $(X_1, Y_1), \ldots, (X_n, Y_n)$, together with a concentration result. For example, one valid test based on Hoeffding's inequality is to reject $H_{0,\lambda}$ if

$$\frac{1}{n} \sum_{i=1}^{n} \text{FDP}(\mathcal{T}_\lambda(X_i), Y_i) + \sqrt{\log(1/\delta)/(2n)} < \alpha. \tag{2}$$

That is, if the empirical risk on the calibration set is far enough below $\alpha$ that we can conclude that the result is not due to chance. With these tests in mind, we do the following. In decreasing order, we test each $\lambda \in \Lambda$ and stop for the first value of $\lambda$ that we fail to reject, i.e., for the largest value of $\lambda$ where (2) fails to hold. Then, we select $\widehat{\lambda}$ to be the preceding value: the smallest value of $\lambda$ considered such that (2) holds. This procedure is valid, as stated next:

**Proposition 1 (Validity of calibration (Angelopoulos et al., 2021))** *With $\widehat{\lambda}$ selected as in Algorithm 1, we have that* (1) *holds.*

This proposition follows from the general result of Learn then Test calibration (Angelopoulos et al., 2021).

### 2.3 Optimizing for Diversity While Controlling the FDR

Often, producing a high-quality recommendation involves more than simply predicting items with high ratings. For example, we may want the set to include a diversity of items—movies of different genres, search results from different sources, and so on—so the user has a more interesting set of options. Loosely, our goal might be to optimize for diversity while maintaining our rigorous FDR guarantee:

$$
\begin{aligned}
\max_{\lambda \in \Lambda} \quad & \mathbb{E}\Big[\text{Diversity}\big(\mathcal{D}_\lambda(X)\big)\Big] \\
\text{s.t.} \quad & \mathbb{P}\Big(\text{FDR}(\mathcal{D}_\lambda) > \alpha\Big) < \delta.
\end{aligned}
\tag{3}
$$

The following technique will work for any diversity measure, although we describe it for one particular choice.

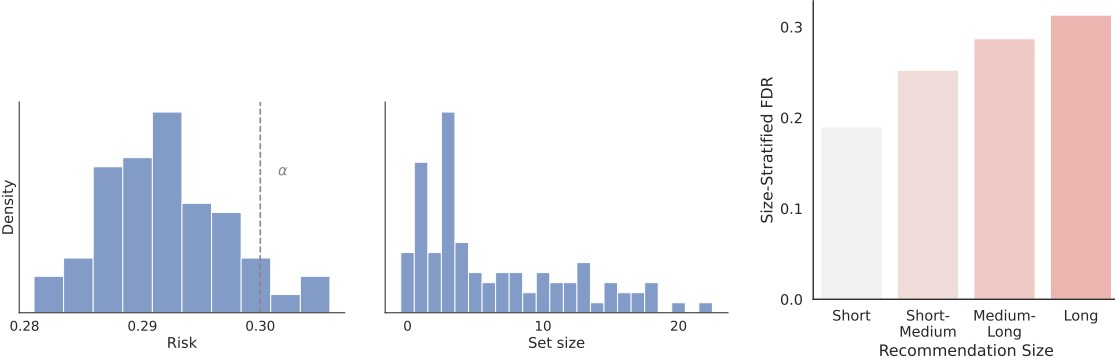

Figure 2: Results on the Yahoo! L2R dataset.

We work in the setting where we wish to recommend no more than $M \in \mathbb{N}$ items to a user due to, for example, constraints on the number of items that can be displayed on a webpage. Furthermore, we assume access to a set of embeddings for each possible response, $E^{(j)} \in \mathbb{R}^d$ for $j \in \{1, ..., K\}$ and $d \in \mathbb{N}$. A natural notion of diversity is based on the average distance among embeddings for responses in a prediction set $S \subseteq \{1, ..., K\}$,

$$\text{Diversity}(S) = \frac{\sum\limits_{j,j' \in S} \left|\left| E^{(j)} - E^{(j')} \right|\right|_2}{\max\left(M, |S|\right)}.$$

The diversity is a deterministic function that takes as input a set of responses $S$ and computes a positive real-valued number: the average distance between elements in $S$. It grows when the embeddings of the elements in $S$ are far apart from each other. Furthermore, the diversity grows when we add more elements until we reach a set size of $M$, after which only the average distance matters. This particular choice of a measure for diversity is not critical; any other metric could be substituted in while maintaining the error-control guarantees below.

Next, we pick our family of sets to maximize the diversity at each choice of $\lambda$. This reduces to subsetting the original prediction set $\mathcal{T}_\lambda$ to the top $M$ most diverse items,

$$\mathcal{D}_\lambda(X) = \underset{\substack{S \subseteq \mathcal{T}_\lambda(X) \\ |S| \leq M}}{\arg\max} \ \text{Diversity}(S).$$

In practice, we do not do the full combinatorial search over items in $\mathcal{T}_\lambda(X)$; instead, we do a greedy approximation, removing the elements that contribute least to diversity first; see Algorithm 3.

With this new family of sets, we choose $\widehat{\lambda}$ as before, replacing $\mathcal{T}$ with $\mathcal{D}$ everywhere it appears. We state the calibration procedure explicitly in Algorithm 2. Despite the seeming mathematical complexity of our diversity optimization, we maintain precise control over the FDR, as stated next.

**Proposition 2 (Validity of calibration with approximate diversity optimization)** *Let $\widehat{\lambda}$ be the result of Algorithm 2. With $\mathcal{T}_\lambda$ as the function given by Algorithm 3, we have that the FDR control in* (1) *holds.*

## 3 Experiments

We consider two popular L2R datasets: the Yahoo! Learning to Rank challenge (Chapelle & Chang, 2011) and the MS MARCO document re-ranking challenge (Nguyen et al., 2016). In each, we use a LambdaRank neural network model (Burges, 2010) with an input size of 700, two hidden layers of size 16 and 8, ReLU activations (Nair & Hinton, 2010), and the Adam optimizer (Kingma & Ba, 2014) with its default parameters: a learning rate of 0.001, momentum parameters $\beta = (0.9, 0.999)$, and $\epsilon = 1e - 7$. The evaluation protocol is shared between both datasets, so we describe it here. First, we randomly split the data into disjoint train and validation sets. We then train our model on the training set. Next, we repeat the following procedure 100 times:

---

**Algorithm 2** The Learn then Test calibration procedure for L2R with approximate diversity optimization

**Input:** Calibration data, $(X_i, Y_i)$, $i = 1, \ldots, n$; embeddings for each calibration point $E_i^{(j)}$, $j = 1, \ldots, K$; risk level $\alpha$; error rate $\delta$; underlying predictor $\widehat{\pi}$; step size $d\lambda > 0$.

**Output:** Parameter $\widehat{\lambda}$ for computing RCPS.

1: $\lambda \leftarrow 1$
2: fdp $\leftarrow 1$
3: **while** fdp $\leq \alpha$ **do**
4:     $\lambda \leftarrow \lambda - d\lambda$
5:     **for** $i = 1, \ldots, n$ **do**
6:         $\mathcal{D}_\lambda(X_i) \leftarrow \mathcal{T}_\lambda(X_i)$
7:         **while** $|\mathcal{D}_\lambda(X_i)| > M$ **do**
8:             leastDiverse $\leftarrow \mathcal{D}_\lambda(X_i)_1$
9:             leastDiversity $\leftarrow \infty$
10:            **for** $t \in \mathcal{D}_\lambda(X_i)$ **do**
11:                **if** Diversity$\big(\mathcal{D}_\lambda(X_i) \setminus t\big) \leq$ leastDiversity **then**
12:                    leastDiverse $\leftarrow t$
13:                    leastDiversity $\leftarrow$ Diversity$\big(\mathcal{D}_\lambda(X_i) \setminus t\big)$
14:         $\mathcal{D}_\lambda(X_i) \leftarrow \mathcal{D}_\lambda(X_i) \setminus$ leastDiverse
15:         $\text{FDP}_i \leftarrow \text{FDP}\big(\mathcal{D}_\lambda(X_i), Y_i\big)$
16:     fdp $\leftarrow \frac{1}{n} \sum\limits_{i=1}^{n} \text{FDP}_i + \sqrt{\frac{1}{2n} \log \frac{1}{\delta}}$        ▷ Can replace with any valid upper-confidence bound on the risk.
17: $\widehat{\lambda} \leftarrow \lambda + d\lambda$                             ▷ Backtrack by one because we overshot.

---

**Algorithm 3** Producing calibrated predictions on a new test-point with diversity optimization in L2R

**Input:** Calibrated parameter $\widehat{\lambda}$; underlying predictor $\widehat{\pi}$; fresh test point $(X, Y)$; maximum number of recommendations $M$.

**Output:** RCPS $\mathcal{D}_\lambda(X)$.

1: $\mathcal{D}_{\widehat{\lambda}}(X) \leftarrow \mathcal{T}_{\widehat{\lambda}}(X)$
2: **while** $|\mathcal{D}_{\widehat{\lambda}}(X)| > M$ **do**
3:     leastDiverse $\leftarrow \mathcal{D}_{\widehat{\lambda}}(X)_1$
4:     leastDiversity $\leftarrow \infty$
5:     **for** $t \in \mathcal{D}_{\widehat{\lambda}}(X)$ **do**
6:         **if** Diversity$\big(\mathcal{D}_{\widehat{\lambda}}(X) \setminus t\big) \leq$ leastDiversity **then**
7:             leastDiverse $\leftarrow t$
8:             leastDiversity $\leftarrow$ Diversity$\big(\mathcal{D}_{\widehat{\lambda}}(X) \setminus t\big)$
9:     $\mathcal{D}_{\widehat{\lambda}}(X) \leftarrow \mathcal{D}_{\widehat{\lambda}}(X) \setminus$ leastDiverse

---

1. Split the validation set into a calibration set and a test set.

2. Using the calibration set, compute $\widehat{\lambda}$ as in Proposition 1.

3. Using the test set, and setting $\lambda$ equal to $\widehat{\lambda}$ wherever it appears, compute the FDR risk as in 2.2, and the set size, $|\mathcal{T}_{\widehat{\lambda}}(X)|$, for a uniformly random choice of $X$ in the validation set.

After this procedure, we have 100 values of the risk and set size, each of which was taken from a different random split of the calibration and test data. We report these values as histograms. Since we know (1) holds, we expect that the histogram of risks will not exceed the chosen value of $\alpha$ except with probability $\delta$; this is indeed the case in our experiments, and the procedure is also not conservative.

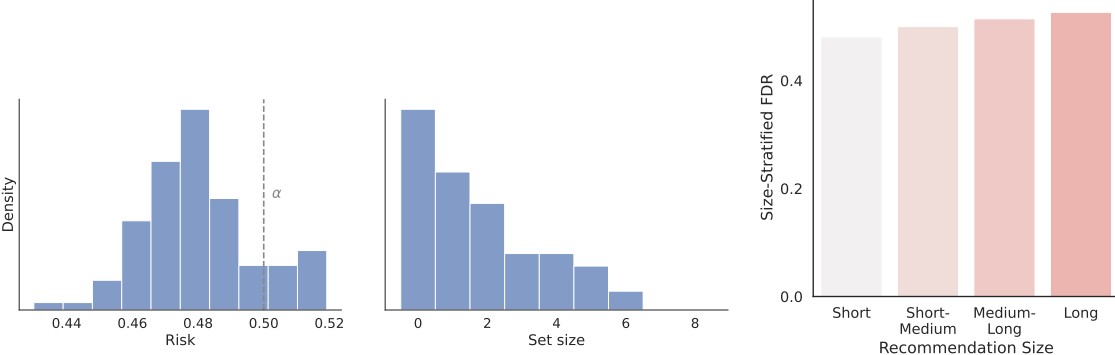

Figure 3: Results on the MS Marco Dataset.

In addition to providing statistical coverage, risk-controlling sets of recommendations should preferably not make systematic errors. One way to check this is by stratifying the risk along some axis and checking the risk within each stratum. If the risk is equal within each stratum, that means systematic errors are not being made along the axis of stratification. We choose to stratify along a generally applicable axis, namely set size. On the validation set, we compute the prediction sets and then compute the risk within each quartile of the set sizes. That is, letting $\left\{ \left( X_i^{(\text{val})}, Y_i^{(\text{val})} \right) \right\}_{i=1}^{n'}$ be the validation examples, we form the bins

$$B_j = \left[ \text{Quantile}\left( \left\{ \left| \mathcal{T}_{\widehat{\lambda}}\left( X_i^{(\text{val})} \right) \right| \right\}_{i=1}^{n'}, \frac{j-1}{4} \right), \text{Quantile}\left( \left\{ \left| \mathcal{T}_{\widehat{\lambda}}\left( X_i^{(\text{val})} \right) \right| \right\}_{i=1}^{n'}, \frac{j}{4} \right) \right],$$

for $j \in \{1, 2, 3, 4\}$. Then, we calculate the empirical FDR within each bin, i.e.,

$$\widehat{\text{FDR}}_j = \frac{\sum\limits_{i=1}^{n'} \text{FDP}\left( \mathcal{T}_{\widehat{\lambda}}\left( X_i^{(\text{val})} \right), Y_i^{(\text{val})} \right) \mathbb{1}\left\{ \left| \mathcal{T}_{\widehat{\lambda}}\left( X_i^{(\text{val})} \right) \right| \in B_j \right\}}{\sum\limits_{i=1}^{n'} \mathbb{1}\left\{ \left| \mathcal{T}_{\widehat{\lambda}}\left( X_i^{(\text{val})} \right) \right| \in B_j \right\}}.$$

We report each of the four stratified risks as a bar in a barplot, labeled 'Short,' 'Short-Medium,' 'Medium-Long,' and 'Long' respectively.

## 3.1 Yahoo! Learning to Rank

The Yahoo! Learning to Rank dataset (Chapelle & Chang, 2011) contains 36251 Yahoo! search queries, where the $i$th query comprises an anonymized embedding vector for each website $X_i^{(j)}$ and the ranking of the $j$th website, $Y_i^{(j)}$, for $j = 1, ..., K$. We subset the dataset to a smaller version with only 26090 queries and use 13045 random points for model training, $n = 8000$ for LTT calibration, and 5045 for testing.

We report the results of the FDR calibration procedure (Algorithm 1) in Figure 2, where we seek FDR control at level $\alpha = 30\%$ with a $\delta = 10\%$ tolerance level. We find that the risk is controlled and that it is nearly tight. Moreover, there is a large spread in the size of the returned set, indicating that the model is effectively discriminating between high-uncertainty and low-uncertainty inputs.

## 3.2 MS Marco

The MS Marco document re-ranking dataset (Nguyen et al., 2016) consists of 367013 text queries sampled from Bing. Each query has 100 documents associated with it, with each document consisting of a title and a body. This task emulates the common real-world scenario where an expressive ranking model must order documents provided to it by a lightweight nominator model. We convert each document and query into a 768-dimensional vector by passing them through a DistilBERT pre-trained model (Sanh et al., 2019) (distilbert-base-uncased in the HuggingFace library (Wolf et al., 2019)). We then concatenate the query and

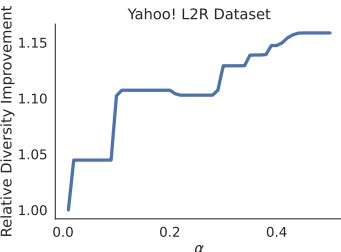 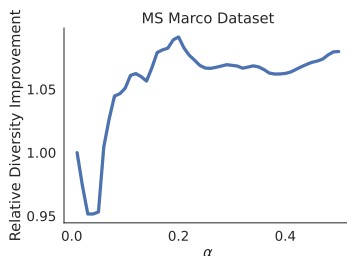

Figure 4: Diversity improvements as a function of $\alpha$ when optimizing for diversity subject to FDR control.

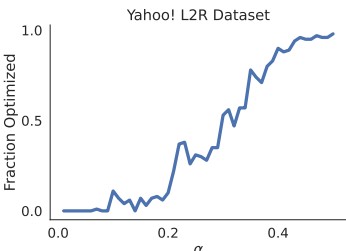 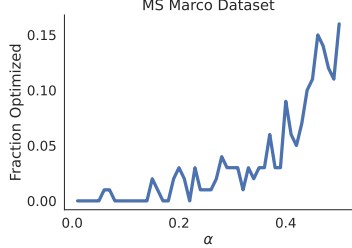

Figure 5: Fraction of elements selected for diversity optimization.

document vectors together for each document-query pair, which we then use as our feature vectors. We subset the dataset to include the first $20,000$ queries and use $10,000$ random points for model training, $n = 1,500$ for LTT calibration, and $8,500$ for testing.

We report the results of the FDR calibration procedure (Algorithm 1) in Figure 3, where we seek FDR control at level $\alpha = 50\%$ with a $\delta = 10\%$ tolerance level. As before, we find that the risk is controlled and that it is nearly tight. We again see a reasonable spread in the size of the returned set, as desired.

### 3.3 Experiments to Optimize Diversity

We also implement experiments on the Yahoo! L2R dataset to understand the properties of the diversity optimization procedure outlined in Section 2.3. As suggested by the optimization problem in (3), there is the diversity of the final set trades off with the stringency of the risk-control guarantee. For a sufficiently loose choice of $\alpha$, the optimal strategy is simply to pick the $M$ most diverse responses out of the $K$ possible ones. As we tighten $\alpha$, the ability to tolerate suboptimal responses decreases, $\mathcal{T}_\lambda$ shrinks, and the diversity optimization has fewer (but higher quality) responses to choose from.

We characterize this tradeoff by sweeping $\alpha$, and for each value, estimating the relative diversity improvement

$$\mathbb{E}\left[\frac{\text{Diversity}\left(\mathcal{D}_{\widehat{\lambda}}(X)\right)}{\text{Diversity}\left(\mathcal{T}_{\widehat{\lambda}}(X)\right)}\right]. \tag{4}$$

More concretely, we repeated the following procedure 100 times.

1. Split the validation set into a calibration set and a test set.

2. Using the calibration set, compute $\widehat{\lambda}$ as in Algorithm 2.

3. Using the test set, and setting $\lambda$ equal to $\widehat{\lambda}$ wherever it appears, compute the FDR risk as in 2.2, and the set size, $|\mathcal{D}_{\widehat{\lambda}}(X)|$, for a uniformly random choice of $X$ in the validation set.

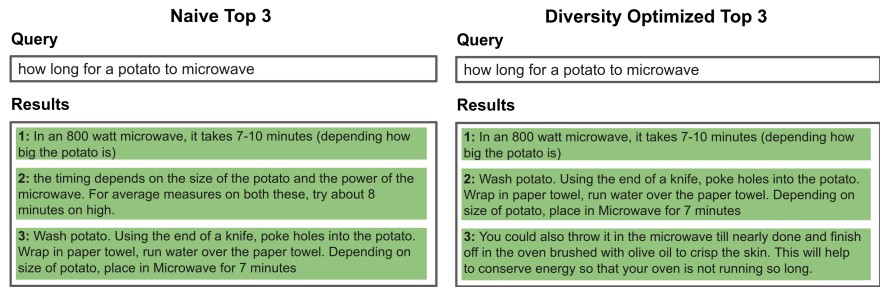

Figure 6: Comparison of the model's top three predictions against those optimized for diversity.

4. Compute and store the values of $\text{Diversity}\big(\mathcal{T}_{\widehat{\lambda}}(X)\big)$ (the diversity before optimization) and $\text{Diversity}\big(\mathcal{D}_{\widehat{\lambda}}(X)\big)$ (the diversity after optimization), as well as the number of sets modified by the the diversity procedure (i.e., those such that $|\mathcal{T}_{\widehat{\lambda}}(X)| > M$).

We estimated (4) by averaging the ratio of the diversities for each set modified by the procedure. The results are shown in Figure 4 for both datasets. The non-monotone fluctuations in the curves are a consequence both of statistical noise and also of our greedy diversity optimization procedure, which does not always pick the optimal set. We also plot the fraction of sets chosen for diversity optimization, i.e., where $\mathcal{T}_{\widehat{\lambda}} > M$, as a function of $\alpha$ in Figure 5. More permissive choices of $\alpha$ allow us to optimize a larger fraction of the sets. An example of a set before and after diversity optimization is shown in Figure 6.

To query how the diversity optimization changed our previous results, we plot the risk and set size on the Yahoo! L2R dataset. The desired FDR level is $\alpha = 30\%$, with a tolerance level of $\delta = 10\%$ and $M = 3$. The plots in Figure 8 indicate the risk is still controlled and the sets have a spread, with most sets being of size three. This is not surprising, since $M = 3$, and all sets whose size was once greater than three are collapsed to size three after diversity optimization. Running the diversity optimization increased the average diversity by 15%, and 40% of the prediction sets were modified by the diversity optimization.

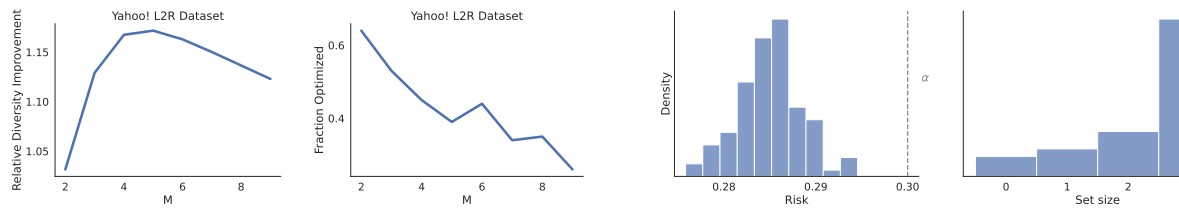

Figure 7: Diversity improvements as a function of $M$ when optimizing diversity subject to FDR control.

Figure 8: The risk and set size on the Yahoo! L2R dataset with diversity optimization, setting $M = 3$.

As a last experiment, we sweep the value of $M$ to see how it affects the diversity of the resulting set. Keeping $\alpha = 0.3$ and $\delta = 0.1$, we vary $M \in \{2, 3, 4, 5, 6, 7, 8, 9\}$ and then estimate the relative diversity improvement in (4) with the same procedure as earlier. As before, we also report the fraction of sets changed by the optimization procedure. See the results plotted in Figure 7 for the Yahoo! L2R dataset. Increasing $M$ excludes smaller sets (ones where the diversity optimization procedure has little room for improvement) from the diversity optimization, while also making the procedure select less aggressively for large sets.

## 4 Discussion

The control of error rates is a critical aspect of the robust, reliable, and trustworthy deployment of learning algorithms. A key stepping stone to meeting such desiderata is to provide rigorous uncertainty quantification for a wide range of loss functions. Providing expressive uncertainty quantification also opens the door to new

algorithmic capabilities. We take a step in this direction for the learning-to-rank problem, showing how to calibrate any base learning algorithm to return sets of items that control the false discovery rate. Further, we show how tracking uncertainty internally allows us to optimize for item diversity, while ensuring that the sets we return have high utility.

We wish to highlight that the framework we leverage here is entirely modular; other components can be grafted in to handle variations of the tasks we consider here. For example, the FDR notion of statistical error can be replaced with others such as the false negative rate, with minimal change to the calibration algorithm. Secondly, we could seek good performance on axes of performance other than set diversity by swapping in another performance criterion in (3). Going even farther, we could optimize for another quantify while requiring that *both* FDR and diversity are maintained at some level. Many recommender system tasks can be handled—with finite-sample statistical guarantees—in the distribution-free risk-control framework.

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
