# OpenReview forum: "Recommendation Systems with Distribution-Free Reliability Guarantees"
_TMLR — Rejected by TMLR_

### Review · Reviewer_qDwG · 2022-07-25

**Summary Of Contributions:**

The paper proposes an algorithm to generate diverse recommendation sets while maintaining a guarantee of minimum quality of service. The algorithm uses the Learn then Test approach to risk control of Angelopoulos et al. 2021, combined with a greedy inference procedure to diversify the recommendations. The authors present some experimental results which attest that indeed the false discovery rate is properly controlled.

**Requested Changes:**

- clarify the formalization of the paper and make correct equations
- discuss how far from optimality the algorithm is, with a preference for a theoretical guarantee if possible
- while it is necessary to show that FDR is controlled, the experiments should provide compelling arguments that the paper does a good job at diversifying results compared to other approaches.
1- for small M, a topline could be an exhaustive search on subsets of size M of T_hat{lambda} to see how much is lost by the greedy algorithm
2- a comparison with maximum marginal relevance (Carbonell et al., 1998), where MMR is run for different values of M, and we report the (observed) FDR and diversity, and compare with those obtained by the current algorithm. While MMR does not provide any guarantee on the FDR, it would be interesting to see how much is gained/lost by the author's approach compared to MMR


**Strengths And Weaknesses:**

strength: the idea of maximizing diversity while controling for FDR is interesting.
weaknesses:
1- many of the equations are incorrect, there is no guarantee on the quality of the diversified set
2- as it stands, the experiments mostly serve the illustrative purpose of FDR control, which is not really the point of the paper since 1) there is a theoretical guarantee on the FDR already, which was proved in previous work, and 2) the point of the paper is not to show that FDR can be controlled (this was already known) but to show how to maximize diversity while controling for FDR. The paper falls short on assessing the quality of the resulting set in terms of diversity.

details:

- the equation for false discovery rate is incorrect. The authors use card(hatS intersect Y*) in the numerator, which corresponds to correct predictions rather than incorrect ones. In particular

- in section 2.1 hat(pi)_{ij} = P(Y^(i) <= Y^(j)) is "an estimate of the probability that item i is preferred to item j", which suggests that Y^(i) is the rank of item i, and small Y^(i) is better. However, in section 2.2 the definition of "False discovery proportions" uses Y^(m+1:K) which suggests that the superscript in Y are not item indices but rather ranks. Moreover, the authors say "take m=.2 K, the top 20% of items" does not make sense: Y^(m+1:K) with m =.2 K would be 80% of the items (top 20% would either be Y^(1:m) or Y^(K-m:K) depending on the convention used (the convention used in the paper is not clear).

- the algorithm guarantees that the FDR is controlled, but it does not guarantee that we maximize the diversity at this level of FDR (because of the indirect maximization over the set T_lambda rather than over all sets that have FDR <= alpha). Even if it is expected that we only obtain an approximation, I would expect some guarantee on the quality of the approximation to support the claim that the algorithm "maximizes" diversity under constraints on the FDR

---

> ### Author Response · Authors · 2022-08-29
> **Response to reviewer qDwG**
>
> Thank you very much for taking the time to read our paper and for your detailed comments. The next revision of our paper will include revisions based on the comments below.
>
> * Typos in equations: thank you for spotting this, we will fix this in our next revision.
> * Diversity optimization doesn't necessarily find optima for a given FDR: We agree this could be made clearer and will rephrase our statements to state that we "approximately maximize" rather than "maximize" diversity in the next revision.  We will also clarify that when we say "approximately maximize", we mean that we are optimizing subject to a fixed model.
> * Comparing diversity experiments to baseline: we would be happy to include an experiment that compares our diversity results with MMR in the next revision of the paper.

---

### Review · Reviewer_L7Wd · 2022-08-08

**Summary Of Contributions:**

The paper introduces a greedy algorithm for selecting items for recommendation tasks which maximize diversity subject to constraints on the false discovery rate. The algorithm, which operates on the output of e.g., learning-to-rank methods, is claimed to be sound based on existing theoretical results. It is evaluated in experiments on two real-world data sets.

**Requested Changes:**

**Main changes**
* In its current state, I view this paper as being on the border of the scope of TMLR. This will be difficult to change. I'd be happy to revise my opinion if the editors think otherwise.

* The work must be better placed in the context of existing work on recommendation systems which account for uncertainty in learning-to-rank, even if they are specialised or make strong assumptions (see top p.3). This includes general writing and motivation, as well as empirical evaluation and baselines.

* Proposition 2 should be justified.

**Typos, minor corrections**

* Figure captions should be much more descriptive (Fig 3 for example does not describe the different plots).

* In the learning-to-rank setting, is the FDR/FDP an appropriate measure? No reference for this is given and the metric ignores the internal rank of the returned items.

* FDR is overloaded on p.4 by a definition which is sensitive to \lambda. The original definition has no such notion. In fact, the original definition on page 2 is agnostic to the rank among items.

* Is there a typo in the stated definition of FDR (Page 2)? $\hat{S} \cap Y^*$ is the subset of items that are both recommended and acceptable, is it not? At least, it seems at odds with the definition of FDP on page 4.

* Eq (3) is just a restatement of the definition on p.2 with a specific class of $\mathcal{D}$

* I would avoid using the notation $S_K$ for the set of permutations, since $S$ is used for the returned items.

* On page 4, $Y^{(i)}$ is used for the rank of item $i$, but previously $Y^*$ was used for the set of acceptable items. This similarity in notation for items in different domains is unfortunate.


**Strengths And Weaknesses:**

## Strengths:

The proposed method is sound and appears to be a useful step in applying learning-to-rank systems with more control in practice.

The related-work section gives a decent overview of the relevant literature in uncertainty estimation. (Although other aspects are missing, see below)

The algorithms appear practical to implement and use.

The evaluation tasks are well chosen.

## Weaknesses:

### Scope

While the primary goal of this work is to post-process the output of machine learning algorithms, learning itself is outside the scope of this paper. In my view, the contributions are only tangentially related to TMLRs stated objective. As such, the paper may be better placed in a venue dedicated to e.g., information retrieval, or more specifically to recommendation systems.

### Technical contribution

The main technical contributions are algorithms for post-processing the output of a probabilistic ranker to select items with optimal diversity and given FDR (Algorithm 2, 3). Algorithm 3 is a greedy algorithm stated without approximation guarantees. This begs multiple questions:
* How bad can the approximation be?
* Are there instances where the full problem can be solved?
* Is there a better approximation algorithm?

Two propositions are used to argue that the proposed algorithms are valid. The proof of Proposition 1 relies solely on Angelopoulos et al. (2021) and Proposition 2 is never proven.
* Is it also unaffected by multiple testing?

### Empirical evaluation

- No baselines are compared to in the empirical evaluation. I am not an expert in recommendation systems, but I find it hard to believe that there are no other approaches to selecting diverse results for recommendation. The related work section does not dive deep into this issue but mentions a few precious works in uncertainty quantification for information retrieval systems.

- The one qualitative example of diversity optimisation is Figure 6, which shows that the idea results in changing a single item. It would have been interesting to see more examples of how diversity optimization improves results, especially when more items are returned.

### Structure and writing

The paper describes it's goal and contributions in several different ways:
* "develop a method for *quantifying uncertainty* for the task of learning to rank"
* "propose a *calibration algorithm* that returns set of items with FDR guaranteed to be lower than a user-specified level with high probability"
* [enable] ... "principled integration of multiple objectives"
* "Distribution-Free Reliability Guarantees"
* "maximize the diversity" [of recommendations]
* "strong guarantees provided by uncertainty quantification"

Concretely, the paper presents an algorithm to maximize recommendation diversity subject to (probabilistic) constraints on the FDR. This is applied post-hoc to an existing ranker. The goal is clearly defined at the end of 1.1. I would advice being as transparent about that goal as possible throughout.

Several of the terms used along the way are debatable choices or poorly described:
* For example, the term uncertainty refers to the (variance/noise in the) recommendation but this is not clearly defined. In the introduction, right before 1.1, an example is given regarding  "misdiagnosis due to uncertain predictions". Such uncertainty could be either aleatoric (noise) or epistemic (model variance/bias) in nature. In this paper, concretely, uncertainty refers to the result in (2), i.e., a concentration result on the average FDP. This is qualitatively quite different from prediction uncertainty since the former is in aggregate and the latter per unit.
* Similarly, the phrase "distribution free" is in the title, but is not explained in the paper.
* On p.1, the authors claim to "develop a method for quantifying uncertainty for the task of learning to rank". However, the method is applied post-hoc to solutions to the learning task. The quote sounds like improvements are made to the learning algorithms, or the analysis thereof.

---

> ### Author Response · Authors · 2022-08-29
> **Response to reviewer L7Wd**
>
> Thank you very much for taking the time to read our paper and for your detailed comments. The next revision of our paper will include revisions based on the comments below.
>
> * Contributions are tangentially related to TMLR: we respectfully disagree with this statement, work on uncertainty quantification has primarily been published at machine learning conferences with a very similar scope to TMLR (e.g. ICML/Neurips). There was, for example, an outstanding paper award at ICML for conformal prediction this year, and several papers in ICML applying it to various domains---our techniques build on a technique similar to conformal prediction. While there do exist conferences/journals focused on information retrieval, work on recommender systems also has a long history of being published in ML conferences, with most of the foundational work on L2R appearing in Neurips/ICML.
> * Proofs of propositions 1 and 2: we agree with the reviewer that these could be made clearer and expanded. We will remedy this in our next revision to include an expanded proof.
> * Baseline comparison in experimental section: as suggested by reviewer qDwG we would be happy to include an experiment that compares our diversity results with MMR in the next revision of the paper.
> * The paper does not actually account for prediction uncertainty: We are indeed doing prediction uncertainty at the per-unit level (where a unit here is a query), not estimating a population-level parameter. The uncertainty we are talking about is both aleatoric and epistemic. The methods we use do not distinguish between the two, and average across all randomness. When we say uncertainty, we do not simply mean the finite sample uncertainty in the estimation of the FDP or the concentration bound; we are talking about providing a predictive reliability guarantee on the model output while using the model's internal notion of uncertainty as a tool. We will include an expanded explanation of this in the introduction, which also further clarifies how our methods are distribution-free. Our techniques are quite similar to conformal prediction or risk-controlling prediction sets, the latter of which provides predictive uncertainty using concentration as a substep.
> * Figure captions should be more descriptive: we agree that the figures could be made clearer with expanded descriptions. We will provide more descriptive captions in the next revision.
> * Is FDR/FDP appropriate in the L2R case: our algorithms do not ignore the ranking of the items. If an item is ranked more highly by the underlying ranking algorithm, more highly ranked items will take precedence over items with lower ranks when considering which items to return for a given query.
> * Typos in equations: thank you for spotting this, we will fix this in our next revision.

---

### Review · Reviewer_PWsn · 2022-08-14

**Summary Of Contributions:**

This paper studied risk-control for learning to rank problem, with the goal of guaranteeing low false discovery rate (FDR). The authors applied Learn then Test method to calibrate learning to rank model to rigorously control FDR. Specifically, the authors proposed to use learn then test fo find a value threshold $\hat\lambda$ that guarantees the FDR of returned ranking is smaller than a given risk level $\alpha$ with probability at least $1-\delta$.  Empirical evaluation on two real-world datasets showed that the proposed methods could control FDR as well as optimize diversity under FDR control.



**Broader Impact Concerns:**

There are no broader impact concerns.

**Requested Changes:**

1. Could the framework be extended to non-binary relevance setting and controlling NDCG? If yes, the authors may include new experiments using the same datasets.
2. Please provide more details on current experiment setting and result discussions.
3. Please provide more explanation on Proposition 1 and 2.

**Strengths And Weaknesses:**


Strengths:

  1. Risk control is a well-motivated task for recommender systems. The main novelty of this paper is applying the recent Learn-then-Test framework to learning to rank problem and achieving rigorous risk-control guarantees. The authors also showed that the framework could be easily extended to diversity optimization under risk control, which is a novel example not discussed in orignial Learn-then-Test paper.

  2. Empirical results on Yahoo! Learning to Rank and MS Marco datasets showed that the proposed method can successfully control FDR under the required level.

  3. The paper is generally well-written and easy to follow. The only concern in writing is missing some important details (see weaknesses).

Weaknesses and Questions:

  1. The major concern is that while the paper claimed to focus on learning to rank problem, the paper did not discuss ranking with difference relavance levels (which is the widely-adopted L2R setting nowadays) but is can only handle binary relavance label (more like classification). FDR control for binary relavance label task is almost the same as Section 3 (Example: FDR Control for Multi-Label Classification) in Angelopoulos et al., 2021. If the goal of this paper is indeed studying risk-control for L2R, I think it is important to examine  whether the framework can also be applied to rigorously control metrics like *NDCG* beyond binary relavance.


  2. Experiment description is not clear. This directly follows previous concern on binary relevance label. In original Yahoo! Learning to Rank dataset, each (query, document) pair is associated with relevance label 0-4. Did the authors generate binary label based on original relevance, e.g., separate by 2? Same detail is missing in evaluation on MS Marco dataset.

  3. Proposition 1 and 2 need detailed explanations. The authors only mentioned that ``This proposition follows from the general result of Learn then Test calibration (Angelopoulos et al., 2021).'', which is vague and can be confusing to readers who are not familiar with Learn the Test paper. I would suggest the authors provide more details on the proof, or at least point to an exact location of the referred paper if the propositions are just re-stating existing results.

Other comments and Typos:
 - FDR definition in Section 1.1 and FDP definition in section 2.2: Instead of $\cap$, the operation should be $\setminus$. Current definition is precision, not FDR.
 - Why do we prefer a wide spread in the size of the returned set as discussed for the second figures in Figure 2 & 3? Also we can observe that the set size is mostly 0 in the histogram on MS Marco; does that mean the ranker returned nothing in order to satisfy $\alpha=30%$? The discussion on these graphs is not clear.
 - Since optimizing diversity is a downstream task of risk control, can we simply apply Algorithm 1 instead of Algorithm 2 and then run Algorithm 3? Would it still be correct (maybe just too conservative)?

Reference:
- Anastasios N. Angelopoulos, Stephen Bates, Emmanuel J. Candès, Michael I. Jordan, and Lihua Lei. Learn then test: Calibrating predictive algorithms to achieve risk control. arXiv preprint arXiv:2110.01052, 2021.

---

> ### Author Response · Authors · 2022-08-29
> **Response to reviewer PWsn**
>
> Thank you very much for taking the time to read our paper and for your detailed comments. The next revision of our paper will include revisions based on the comments below.
>
> * Binary relevance vs. relevance levels: we note that our paper already accounts for multiple relevance levels since our framework allows us to ask for the expected relevance of returned items to be greater than x (with high probability) for any relevance value of x, or for the returned items to be in the top k (with high probability) for any value of k.
> * Experimental details on relevance cutoffs: in our experiments, we chose to define a “good” item as an item with a relevance score that puts it in the top 10 items for that query. We will include these details in the next revision.
> Proofs of propositions 1 and 2: we agree with the reviewer that these could be made clearer and might be confusing to readers not familiar with the literature on learn-then-test. We will remedy this in our next revision to include an expanded proof.
> * Typos in definitions of FDR and FDP: thank you for spotting this, we will fix this in our next revision.
> * Why wide spreads of set sizes are preferable:  A larger spread indicates a form of "dynamic range" in the uncertainties --- if there is a spread of set sizes, then the procedure is distinguishing between low-certainty and high uncertainty cases. We can capture this signal in order to account for model uncertainty while controlling FDR. Consider the extreme case where all sets are of the same size, in that case, we would be independent of how uncertain our model is about different queries---this would be bad.
> * Zero set sizes in MS Marco: returned queries of size 0 imply that the underlying L2R model is highly uncertain and is unable to distinguish between different item qualities. It is essentially returning "I don't know." In practice, we might treat such queries with greater caution, in some scenarios we might still return results to the user but we might signpost that the model is highly uncertain, in other scenarios we might simply return a set of items that are broadly popular.  Note that this is a property of the underlying model, and not our statistical procedure; a better model will output fewer size 0 sets.
> * Unclear descriptions of figures: we agree that the figures could be made clearer with expanded descriptions. We will provide more descriptive captions in the next revision.

---

### Decision · Action_Editors · 2022-09-22

**Recommendation:** Reject

**Comment:**

As pointed on the guidelines, two axes matter for the evaluation of submissions:

Are the claims made in the submission supported by accurate, convincing and clear evidence? The claim that the algorithm "maximizes diversity" sounds like an overstatement as it is only "approximated" under constraints on FPR should be more substantiated with a theoretical analysis or clarification that the approach is heuristic and does not come with any optimality guarantee. The experiments do not compare to baseline techniques aiming to maximize diversity. The paper contains many typos, particularly in the core definitions of basic notions such as FDR and many equations, as pointed out by qDwG and Pwsn. Proofreading should not rely on reviewers, and a cleaner version is needed.

Would some individuals in TMLR's audience be interested in the findings of this paper? In its current state, the paper only presents an idea without profoundly studying it either from a theoretical or an empirical standpoint, reducing the community's interest. All reviewers mentioned this concern which is perfectly normal for a workshop paper but is not acceptable for a journal version. TMLR aims to contribute to the understanding of the computational and mathematical principles that enable intelligence through learning, be it in brains or in machines. As pointed by L7Wd the proposition is post-processing the results of a learning procedure to maximize diversity under FDR constraints. Yet attractive in some domain-specific applications such as recommender systems, the interest for TMLR goals is borderline.

Indeed under a major revision, my opinion could change.